# Ambient air pollution and cause-specific risk of hospital admission in China: A nationwide time-series study

**Jiangshao Gu** [1,2,3]*, **Ying Shi**[4], **Yifang Zhu** [5,6], **Ning Chen**[1,2,3], **Haibo Wang** [4,7], **Zongjiu Zhang**[8‡], **Ting Chen** [1,2,3‡]*

**1** Center for Big Data Research in Health and Medicine, Institute for Data Sciences, Tsinghua University, Beijing, China, **2** Tsinghua-Fuzhou Institute of Digital Technology, Beijing National Research Center for Information Science and Technology, Tsinghua University, Beijing, China, **3** Institute for Artificial Intelligence, State Key Lab of Intelligent Technology and Systems, Department of Computer Science and Technology, Tsinghua University, Beijing, China, **4** China Standard Medical Information Research Center, Shenzhen, China, **5** Department of Environmental Health Science, University of California at Los Angeles, Los Angeles, California, United States of America, **6** Institute of the Environment and Sustainability, University of California at Los Angeles, Los Angeles, California, United States of America, **7** Clinical Trial Unit, First Affiliated Hospital of Sun Yat-Sen University, Guangzhou, China, **8** Bureau of Medical Administration, National Health Commission of the People's Republic of China, Beijing, China

‡ These authors are joint senior authors on this work.
* gujs15@mails.tsinghua.edu.cn (JG); tingchen@tsinghua.edu.cn (TC)

**Data Availability Statement:** Air pollution data can be accessed from the National Air Pollution Monitoring System (http://106.37.208.233:20035/ ). Meteorological data can be accessed from the

## Abstract

### Background

The impacts of air pollution on circulatory and respiratory systems have been extensively studied. However, the associations between air pollution exposure and the risk of noncommunicable diseases of other organ systems, including diseases of the digestive, musculoskeletal, and genitourinary systems, remain unclear or inconclusive. We aimed to systematically assess the associations between short-term exposure to main air pollutants (fine particulate matter [$PM_{2.5}$] and ozone) and cause-specific risk of hospital admission in China over a wide spectrum of human diseases.

### Methods and findings

Daily data on hospital admissions for primary diagnosis of 14 major and 188 minor disease categories in 252 Chinese cities (107 cities in North China and 145 cities in South China) from January 1, 2013, to December 31, 2017, were obtained from the Hospital Quality Monitoring System of China (covering 387 hospitals in North China and 614 hospitals in South China). We applied a 2-stage analytic approach to assess the associations between air pollution and daily hospital admissions. City-specific associations were estimated with quasi-Poisson regression models and then pooled by random-effects meta-analyses. Each disease category was analyzed separately, and the *P* values were adjusted for multiple comparisons. A total of 117,338,867 hospital admissions were recorded in the study period. Overall, 51.7% of the hospitalized cases were male, and 71.3% were aged <65 years. Robust positive associations were found between short-term $PM_{2.5}$ exposure and hospital

China Meteorological Data Sharing Service System (http://data.cma.cn/). Hospital admission data were obtained from the Hospital Quality Monitoring System (HQMS) (https://www.hqms.org.cn/) and cannot be made publicly available due to ethical and legal restrictions. These routinely collected healthcare data, though anonymized and de-identified, contain potentially identifying or sensitive patient information. Data that are not directly identifying, including the date of admission, clinical diagnosis, and demographic information, can become identifying in combination. According to the Personal Information Protection Law in the People's Republic of China, these data cannot be shared publicly. The data are available upon request for researchers who meet the criteria for access to confidential data. Data requesters are required to submit a research proposal to the scientific committee of the HQMS via the Scientific Project Management System (https://spms.hqms.org.cn/), under the regulation of Bureau of Medical Administration, National Health Commission of the People's Republic of China. The research proposal should address: (1) the background and rationale of the proposed research project; (2) the scope of the relevant data (eg, the list of relevant data items, time range, hospital scope, and the list of relevant ICD-10 codes); (3) the significance of the expected results in terms of informing public health policy makers. The scientific committee of the HQMS will review the research proposal to ensure the appropriateness of its intended use, and the data will be available if the proposal is approved. For more information and for technical assistance, please contact the HQMS staff (service@hqms.org.cn).

**Funding:** JG, NC, and TC were supported by the National Natural Science Foundation of China (grants 61872218, 61721003, and 61673241; http://www.nsfc.gov.cn) and Beijing National Research Center for Information Science and Technology (BNRist). The funders had no role in study design, data collection and analysis, decision to publish, or preparation of the manuscript.

**Competing interests:** The authors have declared that no competing interests exist.

**Abbreviations:** CCS, Clinical Classifications Software; COPD, chronic obstructive pulmonary disease; df, degrees of freedom; HQMS, Hospital Quality Monitoring System; ICD, 10, International Classification of Diseases, 10th Revision; RR, relative risk.

admissions for 7 major disease categories: (1) endocrine, nutritional, and metabolic diseases; (2) nervous diseases; (3) circulatory diseases; (4) respiratory diseases; (5) digestive diseases; (6) musculoskeletal and connective tissue diseases; and (7) genitourinary diseases. For example, a 10-μg/m$^3$ increase in $PM_{2.5}$ was associated with a 0.21% (95% CI 0.15% to 0.27%; adjusted $P < 0.001$) increase in hospital admissions for diseases of the digestive system on the same day in 2-pollutant models (adjusting for ozone). There were 35 minor disease categories significantly positively associated with same-day $PM_{2.5}$ in both single- and 2-pollutant models, including diabetes mellitus, anemia, intestinal infection, liver diseases, gastrointestinal hemorrhage, renal failure, urinary tract calculus, chronic ulcer of skin, and back problems. The association between short-term ozone exposure and respiratory diseases was robust. No safety threshold in the exposure–response relationships between $PM_{2.5}$ and hospital admissions was observed. The main limitations of the present study included the unavailability of data on personal air pollution exposures.

## Conclusions

In the Chinese population during 2013–2017, short-term exposure to air pollution, especially $PM_{2.5}$, was associated with increased risk of hospitalization for diseases of multiple organ systems, including certain diseases of the digestive, musculoskeletal, and genitourinary systems; many of these associations are important but still not fully recognized. The effect estimates and exposure–response relationships can inform policy making aimed at protecting public health from air pollution in China.

## Author summary

### Why was this study done?

- Besides the well-known cardiorespiratory effects of air pollution, an increasing number of studies suggest that air pollution might be associated with certain non-cardiorespiratory diseases.

- Evidence of associations between air pollution exposure and the risk of many non-cardiorespiratory diseases is still scarce and inconclusive.

- Few studies have characterized the acute health effects of air pollution on multiple organ systems using uniform methodology and databases.

### What did the researchers do and find?

- We conducted a national time-series study using data of 117,338,867 hospital admissions for 14 major and 188 minor disease categories in 252 Chinese cities from 2013 to 2017, to assess the associations between short-term exposure to fine particulate matter ($PM_{2.5}$) and ozone and cause-specific risk of hospital admission on a national scale.

- City-specific associations were estimated with quasi-Poisson regression models and then pooled by random-effects meta-analyses. Each disease category was analyzed separately, and the *P* values were adjusted for multiple comparisons.

- Short-term $PM_{2.5}$ exposure was significantly positively associated with hospital admissions for 13 major disease categories (of which 7 associations were considered robust, including for diseases of the digestive, musculoskeletal, and genitourinary systems) and 35 minor disease categories, whether adjusted for ozone or not. The association between short-term ozone exposure and respiratory diseases was robust.

## What do these findings mean?

- To our knowledge, this is the first national study in China aimed at systematically investigating possible ways in which short-term air pollution exposure may be associated with severe illnesses requiring hospitalization.

- Our findings highlight the extensive adverse impacts of air pollution on human health, and indicate the significant social benefits of effective mitigation measures.

## Introduction

Air pollution has become a public health concern worldwide, with exposure linked to increased morbidity and mortality [1–4]. Criteria air pollutants include fine particulate matter of 2.5 μm or less in aerodynamic diameter ($PM_{2.5}$), tropospheric ozone ($O_3$), sulfur dioxide ($SO_2$), nitrogen dioxide ($NO_2$), and carbon monoxide (CO). In particular, $PM_{2.5}$ and $O_3$ were used by the Global Burden of Disease Study as the 2 indicators to quantify population exposure to air pollution when estimating the global burden of disease attributable to ambient air pollution, since they were the most consistent and robust predictors of adverse outcomes in previous studies [5].

The health effects of air pollution on the circulatory and respiratory systems are well documented [6,7]. There is emerging evidence for increased risk of some non-cardiorespiratory diseases related to air pollution, e.g., diabetes, autism in children, dementia in the elderly, and premature birth and low birthweight [8,9]. Several recent studies also revealed positive associations between air pollution and certain diseases of the digestive, skeletal, and urinary systems, e.g., peptic ulcer bleeding [10], bone loss over time and bone fracture [11], and incident chronic kidney disease and progression to end-stage renal disease [12–14]. There are at least 3 hypotheses to explain the extrapulmonary effects of air pollution. First, inhaled pollutants may induce pulmonary inflammation and oxidative stress, which is sufficient to cause systemic inflammation and oxidative stress [6,15]. Second, the lung autonomic nervous system may be provoked by pulmonary exposure, which could then result in autonomic nervous system imbalance [15]; the levels of stress hormones may also be altered [16]. Third, the pollutants may reach and directly interact with remote organs, e.g., by penetrating the alveoli and entering the circulation [17,18]. However, the epidemiological evidence is still scarce and inconclusive for many diseases, and is usually subject to small data size, limited representativeness, and

possible publication bias. There are few studies that have characterized the health effects of air pollution on multiple organ systems using uniform methodology and databases.

China has a population of more than 1.3 billion being exposed to relatively high levels of air pollution. In recent years, China has gradually built up high-quality national databases for inpatient discharge registration, which, combined with national monitoring networks of ambient air quality, provide an opportunity to systematically investigate possible ways in which air pollution exposure may be associated with severe illnesses requiring hospitalization. In this study, we performed a nationwide time-series analysis based on hospital admissions in 252 Chinese cities from 2013 to 2017, to assess the associations between short-term exposure to main air pollutants ($PM_{2.5}$ and $O_3$) and cause-specific risk of hospital admission on a national scale. Diseases were classified by 2 approaches at different granularities. Potential effect modifiers and exposure–response relationships were also evaluated.

## Methods

### Data collection

The daily hospital admission data during the period January 1, 2013, to December 31, 2017, came from the Hospital Quality Monitoring System (HQMS), a national registration database of electronic inpatient discharge records of class 3 hospitals in China, under the administration of the National Health Commission of the People's Republic of China. Class 3 hospitals are the highest ranked medical institutions in China's healthcare system, corresponding to tertiary hospitals in the US and Europe but also providing primary and secondary care to less serious patients. In Western countries, healthcare resource utilization is generally scheduled by appointment. However, there is no general-practitioner-based referral system in China [19,20]. Regular outpatient visits and hospital admissions are generally unscheduled and are on a first-come, first-served basis. People usually go to hospital promptly when they develop symptoms and will be admitted immediately if necessary. Ninety-five percent of the total Chinese population was covered by social health insurance schemes by the end of 2017 [21]. Therefore, hospital admission records can provide reliable and timely information on the health status of a geographically defined population in China. Since January 1, 2013, class 3 hospitals in China have been mandated to automatically submit inpatient discharge records to the HQMS on a daily basis, in a nationally standardized format. Each record describes a hospital stay using 346 information items, from which we extracted the date of admission, sex, age, and the primary discharge diagnosis. The diagnosis is coded using the International Classification of Diseases, 10th Revision (ICD-10) [22] by certified professional medical coders at each hospital. Quality control is automatically performed at the time of submission to guarantee the completeness, consistency, and accuracy of data. By December 31, 2017, the HQMS covered 1,001 (67.0%) class 3 hospitals and 1,011,375 (78.4%) hospital beds in 252 Chinese cities. The population, the total number of class 3 hospitals, the number of class 3 hospitals covered by the HQMS, the total number of hospital beds, the number of hospital beds covered by the HQMS, and the coverage rates of class 3 hospitals and hospital beds by the HQMS in each of the 252 cities are presented in S1 Table. This database has been used to track the epidemiological situation and trends of several diseases in China [23–25].

To clearly characterize the health effects of air pollution over a wide spectrum of human diseases, we defined the disease categories at 2 levels. Major disease categories are based on the chapter division (first-level classification) of the ICD-10 diagnostic coding system [22]. Specifically, we treated each ICD-10 chapter as a candidate category and assigned each diagnosis code to the ICD-10 chapter that it belongs to. There are 22 ICD-10 chapters in total, of which the first 14 were included in the scope of this study (14 major disease categories). Minor

disease categories are based on the Clinical Classifications Software (CCS), a validated approach developed by the Agency for Healthcare Research and Quality [26]. The CCS is a comprehensive categorization scheme for aggregating diagnosis codes into a manageable number of disease categories on the basis of clinical homogeneity, and has been widely used to examine patterns of specific health conditions [27,28]. All diagnosis codes were collapsed into 260 mutually exclusive CCS categories, of which we selected as candidates 191 that conceptually belonged to the 14 major disease categories. We excluded 3 categories (CCS codes 56, 61, and 75) that had too few data, leaving 188 as the final set of minor disease categories. For each cause (major or minor disease category), we obtained the daily counts of citywide hospital admissions by summing the daily number of admissions for primary diagnosis of this cause in each hospital in a city. All data used were anonymized and de-identified prior to analysis, under the supervision of Bureau of Medical Administration, National Health Commission of the People's Republic of China. Because the data were analyzed at the aggregate level with no individual identifiers involved, institutional review board approval and participant written consent were not required for this study.

The daily air pollution data came from the National Air Pollution Monitoring System (http://106.37.208.233:20035/), administered by China's Ministry of Ecology and Environment. The number of monitoring stations in a city ranges from 1 to 17, with a median of 4. These fixed-site stations are mandated not to be situated in the neighborhood of distinct emission sources (including but not limited to traffic, industry, and open burning). All measurement procedures are in accordance with China's Ambient Air Quality Standards (GB3095-2012). For each city, we derived daily 24-hour average concentrations of $PM_{2.5}$, $SO_2$, $NO_2$, and CO and maximum 8-hour average concentrations of $O_3$, averaged across all valid monitoring sites, to represent the population exposure to ambient air pollution. We also collected daily mean temperature and mean relative humidity for each city from the China Meteorological Data Sharing Service System (http://data.cma.cn/). During the study period, the missing data rates were 1.69% for $PM_{2.5}$, 1.72% for $O_3$, 1.68% for $SO_2$, 1.67% for $NO_2$, 1.72% for CO, 0.70% for temperature, and 0.70% for relative humidity. Days with missing monitoring measurements were excluded from analysis.

## Statistical analysis

We applied a 2-stage analytic approach to assess the associations between short-term exposure to air pollutants and daily hospital admissions; this approach has been widely used in previous multisite time-series studies [1,29–33]. The methods and models used in this study were determined before the analyses were conducted. In the first stage, we fit quasi-Poisson regression models separately to the daily time-series data of air pollutants and hospital admissions in each city for each cause, linked by date, to estimate the city- and cause-specific relative risk (RR) of hospital admission associated with air pollutants. The potential confounding effects of weather, seasonality, and long-term patterns were controlled for by smoothing functions (natural cubic splines). Specifically, we introduced the following covariates in the models, which were prespecified according to previously published studies [1,19,20,29–38]: (1) a natural cubic spline smoother of calendar day with 10 degrees of freedom (*df*) per year; (2) natural cubic spline smoothers of the temperature on the same day as admission (lag 0) and the average temperature over the 3 days before admission (lag 1–3), both with 6 *df*; (3) natural cubic spline smoothers of the relative humidity at lag 0 and lag 1–3, both with 3 *df*; (4) indicator variables for the day of the week and public holidays. The *df* values for calendar day, temperature, and relative humidity were selected based on the parameters used in previously published studies [1,19,20,29–38], and were further examined by sensitivity analyses described below. In

the second stage, the city-specific RR estimates were pooled by random-effects meta-analyses to generate the national average RR estimates and $P$ values testing the null hypothesis of no association [36,39].

For $PM_{2.5}$ and $O_3$, we estimated their effects on daily hospital admissions using single-pollutant models and 2-pollutant models (adjusting for each other), as well as 3-pollutant models additionally adjusting for 1 of $SO_2$, $NO_2$, and CO to examine the robustness of our results. Multiple lag structures of $PM_{2.5}$ and $O_3$ (single-day and moving average exposures up to 3 days before admission) were used to explore the lag patterns in the acute effects of air pollution exposure. We always used the 2-day moving average exposure (lag 0–1) as the exposure metric of co-pollutants in 2- and 3-pollutant models [32,33]. To address the multiple testing problem, we applied the Benjamini–Hochberg procedure to adjust the $P$ values [40]. Specifically, given a pollutant and a model specification (including the lag of exposure, adjustment for co-pollutants, and $df$ for smoothing functions), we could derive 14 and 188 $P$ values for major and minor disease categories, respectively. The 14 or 188 $P$ values were treated as a batch and adjusted together through the Benjamini–Hochberg procedure. The associations with adjusted $P < 0.05$ were considered statistically significant, keeping the false discovery rate $< 5\%$.

To explore potential effect modifiers, we conducted subgroup analyses by sex (male and female), age ($<65$, 65–74, and $\geq 75$ years), season (cool season, from October to March; warm season, from April to September) [3,4], and region (North China and South China, divided by the Huai River–Qinling Mountain line [the 33th parallel of north latitude, in practice]) (S1 Fig) [41]. The statistical significance of effect modification (difference in effect estimates between subgroups) was tested by a 2-sample $Z$-test [3,4]. We did not adjust the $P$ values for association or effect modification in the subgroup analyses due to the exploratory nature of these analyses. Following the suggestion of reviewers, we further divided China into 6 regions based on geography, climate, and culture (middle north, northeast, east, middle south, southwest, and northwest) (S1 Fig), and repeated the main analyses in each region to obtain the regional average estimates. We developed the exposure–response curves of the relationship between air pollution and hospitalization at the national level as was done in previous studies [42,43]. Briefly, we replaced the linear term for air pollutants in the first-stage regression model with a natural cubic spline smoother with 3 $df$ (technically, the coefficient of log RR was replaced by 3 coefficients used to represent the spline, and the variance of the coefficient was replaced by a $3 \times 3$ variance–covariance matrix). We then conducted multivariate random-effects meta-analyses to generate the national average estimates of the exposure–response curves.

As sensitivity analyses, we changed the $df$ for the smoothing function of calendar day over the range of 6 to 14 per year, for temperature over the range of 3 to 9, and for relative humidity over the range of 3 to 9. These ranges were chosen to be consistent with previous nationwide studies [19,20,31,36–38]; we specially tested if the associations remained after more aggressive adjustment for potential confounding effects of time trends and weather conditions (i.e., using more $df$). Following the suggestion of reviewers, we included 8 outcomes (the CCS categories birth trauma; joint disorders and dislocations, trauma-related; spinal cord injury; sprains and strains; poisoning by psychotropic agents; poisoning by other medications and drugs; rehabilitation care, fitting of prostheses, and adjustment of devices; and medical examination/evaluation) as negative controls [44], for which no biological or clinical evidence supports an association with air pollution. We repeated the main analyses on these negative control outcomes to examine if our results were substantially biased owing to residual confounding.

Finally, to gauge the potential public health benefits of air pollution control measures, we calculated the annual reduction in hospital admissions ($H$) and hospitalization expenses ($E$) attributable to a 10-$\mu g/m^3$ reduction in the daily $PM_{2.5}$ level in China [1,20,30,31,37]. $H$ is

defined as $H = (\exp(\beta \times \Delta x) - 1) \times N$, where $\beta$ is the national average effect estimate (log RR) for a 1-μg/m$^3$ change in PM$_{2.5}$ from the main analyses, $\Delta x$ is 10 μg/m$^3$, and $N$ is the total number of hospital admissions in China in 2016, collected from China Health and Family Planning Statistical Yearbook 2017 (S2 Table). $E$ is defined as $E = c \times H$, where $c$ is the average cost for each hospitalization in China in 2016, collected from China Health and Family Planning Statistical Yearbook 2017 (S2 Table). $H$ and $E$ were calculated for each cause and then added up to get an overall estimate.

Results are presented as point estimates and 95% confidence intervals (CIs) of the percentage increase in daily hospital admissions associated with a 10-μg/m$^3$ increase in PM$_{2.5}$ or O$_3$. $P$ values are always 2-sided. All analyses were done in R software version 3.4.4 (R Foundation for Statistical Computing), with package *mgcv* for fitting regression models and package *mvmeta* for conducting random-effects meta-analyses.

## Results

A total of 117,338,867 hospital admissions in 252 Chinese cities (107 cities in North China and 145 cities in South China) (S1 Fig) from 2013 to 2017 were included in this study, collected from 1,001 class 3 hospitals covered by the HQMS (387 hospitals in North China and 614 hospitals in South China) (S1 Table). The demographic characteristics of the health data are provided in Table 1 and S3 Table. Overall, 51.7% of the hospitalized cases were male, and 71.3% were aged <65 years. The distributions of daily counts of citywide hospital admissions are provided in Table 2. On average per city, there were 273.3 hospital admissions per day for 14 major disease categories and 188 minor disease categories. The national average levels of ambient air pollutants were 50.6 μg/m$^3$ for PM$_{2.5}$, 87.2 μg/m$^3$ for O$_3$, 24.0 μg/m$^3$ for SO$_2$, 31.4 μg/m$^3$ for NO$_2$, and 1.1 mg/m$^3$ for CO during the study period. More statistics of the environmental data (distributions of citywide annual-average levels and national average Pearson correlation coefficients) are provided in S4 and S5 Tables.

Generally, the effect estimates for PM$_{2.5}$ for most disease categories were largest on the same day (lag 0 days), and decreased sharply in the subsequent days (S2 Fig; S6–S12 Tables). The lag patterns of the effect estimates for O$_3$ varied by disease category, with same-day exposure (lag 0 days) leading to the largest effect estimates for some diseases (e.g., diseases of the circulatory system) and previous-day exposure (lag 1 day) leading to the largest effect estimates for other diseases (e.g., diseases of the respiratory system) (S3 Fig; S13–S19 Tables). At the

**Table 1. Demographic characteristics of hospital admissions for 14 major disease categories in 252 Chinese cities, 2013–2017.**

| Characteristic | Nationwide | North | South |
|---|---|---|---|
| **Total *n*** | 103,230,193 | 39,376,209 | 63,853,984 |
| **Sex, *n* (%)** | | | |
| Male | 54,034,143 (52.3) | 20,416,992 (51.9) | 33,617,151 (52.6) |
| Female | 49,196,050 (47.7) | 18,959,217 (48.1) | 30,236,833 (47.4) |
| **Age, years, *n* (%)** | | | |
| <65 | 72,508,804 (70.2) | 27,715,976 (70.4) | 44,792,828 (70.1) |
| 65–74 | 16,821,027 (16.3) | 6,538,588 (16.6) | 10,282,439 (16.1) |
| ≥75 | 13,900,362 (13.5) | 5,121,645 (13.0) | 8,778,717 (13.7) |

Data presented here are based on the ICD-10 codes (primary discharge diagnosis codes) covered by the 14 major disease categories. There are additionally 14,108,674 hospital admissions with ICD-10 codes covered by the 188 minor disease categories but not by the 14 major disease categories, because the 2 levels of disease categories form an approximate but not strict hierarchical structure. The 2 regions of China (North and South) are divided by the Huai River–Qinling Mountain line. Demographic characteristics of cause-specific hospital admissions are provided in S3 Table.

**Table 2. Summary statistics of annual-average daily counts of citywide hospital admissions in 252 Chinese cities, 2013–2017.**

| Major disease category | Mean | Standard deviation | Minimum | Percentile | | | | | Maximum |
|---|---|---|---|---|---|---|---|---|---|
| | | | | 10th | 25th | 50th | 75th | 90th | |
| All | 273.3 | 412.4 | 1.7 | 51.2 | 86.8 | 141.4 | 262.0 | 649.8 | 2,909.7 |
| Certain infectious and parasitic diseases | 10.3 | 14.7 | 0.0 | 1.5 | 3.1 | 5.8 | 10.9 | 22.5 | 141.2 |
| Neoplasms | 36.1 | 73.1 | 0.0 | 3.1 | 6.5 | 13.7 | 30.7 | 84.5 | 712.3 |
| Diseases of the blood and blood-forming organs and certain disorders involving the immune mechanism | 3.5 | 4.7 | 0.0 | 0.5 | 1.0 | 2.0 | 3.6 | 7.3 | 29.6 |
| Endocrine, nutritional, and metabolic diseases | 11.6 | 16.8 | 0.0 | 2.2 | 3.8 | 6.2 | 11.0 | 26.3 | 123.1 |
| Mental and behavioral disorders | 3.4 | 6.0 | 0.0 | 0.2 | 0.5 | 1.0 | 3.4 | 10.1 | 40.1 |
| Diseases of the nervous system | 10.4 | 15.6 | 0.1 | 1.4 | 3.1 | 5.5 | 10.3 | 24.9 | 115.5 |
| Diseases of the eye and adnexa | 10.2 | 20.1 | 0.0 | 0.8 | 1.9 | 4.0 | 8.7 | 25.5 | 184.0 |
| Diseases of the ear and mastoid process | 2.8 | 4.2 | 0.0 | 0.3 | 0.8 | 1.5 | 2.7 | 6.9 | 31.7 |
| Diseases of the circulatory system | 47.7 | 68.2 | 0.0 | 9.5 | 16.0 | 27.1 | 49.3 | 103.4 | 515.6 |
| Diseases of the respiratory system | 35.5 | 40.9 | 0.2 | 7.5 | 13.8 | 22.6 | 41.5 | 72.9 | 324.7 |
| Diseases of the digestive system | 31.9 | 40.7 | 0.0 | 6.2 | 11.7 | 18.6 | 34.0 | 68.4 | 294.4 |
| Diseases of the skin and subcutaneous tissue | 3.0 | 5.0 | 0.0 | 0.3 | 0.6 | 1.3 | 3.1 | 7.4 | 45.3 |
| Diseases of the musculoskeletal system and connective tissue | 12.9 | 22.1 | 0.0 | 1.6 | 2.8 | 5.9 | 12.8 | 30.6 | 192.5 |
| Diseases of the genitourinary system | 21.6 | 34.6 | 0.0 | 3.0 | 6.3 | 10.9 | 20.1 | 51.9 | 269.2 |

The hospital admissions are grouped into 14 major disease categories by primary discharge diagnosis codes, based on the chapter division of the ICD-10 diagnostic coding system.

national level, the same-day concentration of $PM_{2.5}$ (lag 0 days) was significantly positively associated with hospital admissions for 13 major disease categories in both single- and 2-pollutant models (Fig 1). For example, each 10-μg/m$^3$ increase in same-day $PM_{2.5}$ was associated with a 0.21% (95% CI 0.15% to 0.27%; adjusted $P < 0.001$) increase in hospital admissions for diseases of the digestive system when adjusted for $O_3$. In particular, the associations of $PM_{2.5}$ with 7 major disease categories were robust to the lag of exposure and further adjustment for co-pollutants (S2 and S4 Figs): (1) endocrine, nutritional, and metabolic diseases; (2) diseases of the nervous system; (3) diseases of the circulatory system; (4) diseases of the respiratory system; (5) diseases of the digestive system; (6) diseases of the musculoskeletal system and connective tissue; and (7) diseases of the genitourinary system. At the national level, the 2-day moving average concentration of $O_3$ (lag 0–1 days) was significantly positively associated with hospital admissions for diseases of the respiratory system in both single- and 2-pollutant models, and the association was robust to the lag of exposure and further adjustment for co-pollutants (Fig 1, S3 and S5 Figs).

The national average RR estimates for $PM_{2.5}$ and $O_3$ by minor disease category are shown in Table 3 and S6–S19 Tables. The same-day concentration of $PM_{2.5}$ (lag 0 days) was significantly positively associated with hospital admissions for 35 minor disease categories in both single- and 2-pollutant models, 6 categories in single-pollutant models only, and 4 categories in 2-pollutant models only. Among the 35 minor disease categories significantly associated with $PM_{2.5}$ in both single- and 2-pollutant models, there were 9 categories of circulatory diseases including essential hypertension, acute myocardial infarction, ischemic heart diseases, pulmonary heart disease, cardiac dysrhythmias, congestive heart failure, acute cerebrovascular disease, and transient cerebral ischemia; 6 categories of respiratory diseases, including pneumonia, acute bronchitis, upper respiratory infections, and chronic obstructive pulmonary disease (COPD) and bronchiectasis; 7 categories of digestive diseases, including intestinal

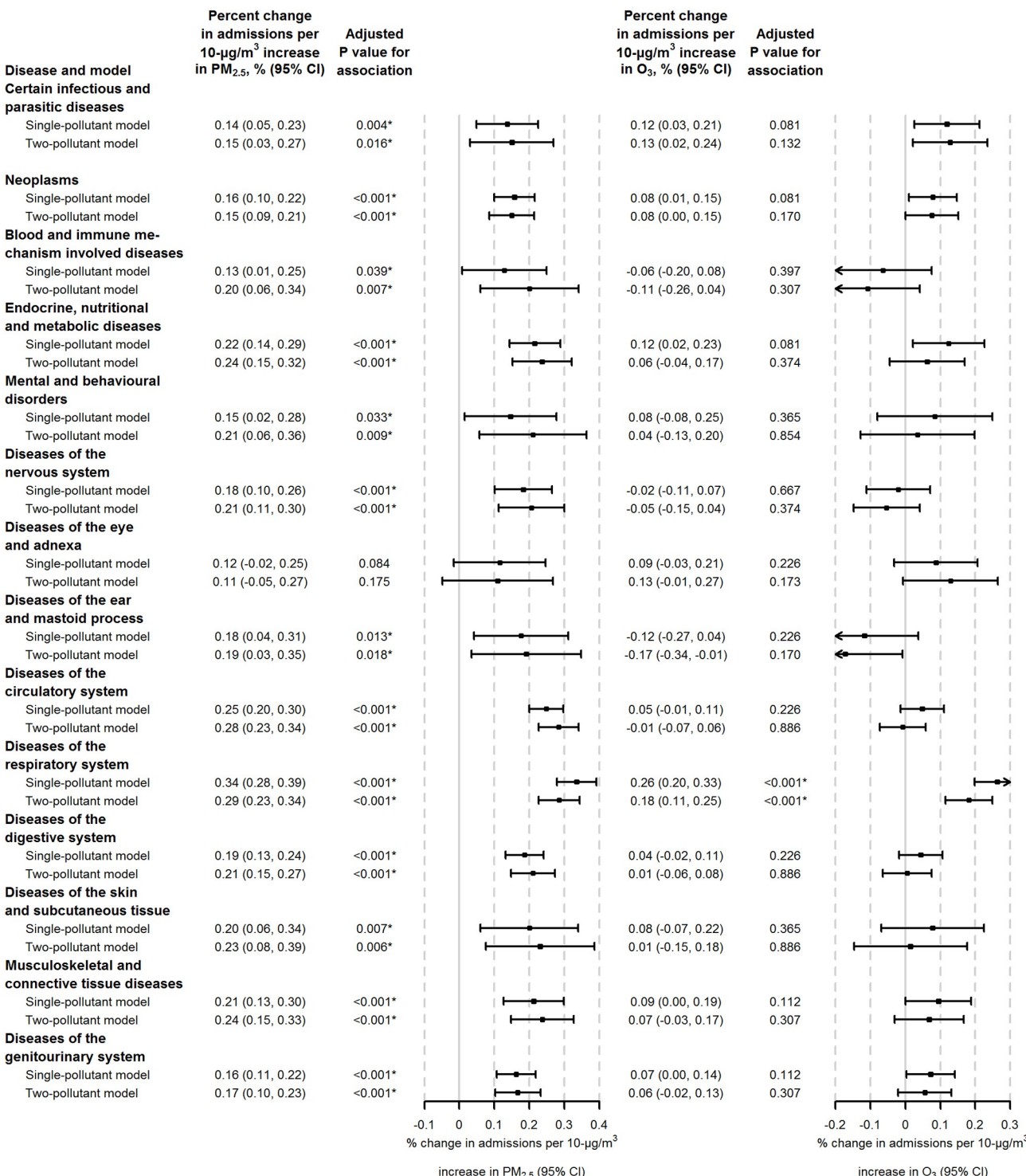

**Fig 1. Percent change in hospital admissions per 10-μg/m³ increase in PM₂.₅ and O₃ by major disease category, on average across all cities.** Results are presented as point estimates and 95% CIs of the percentage increase in daily hospital admissions associated with a 10-μg/m³ increase in PM₂.₅ or O₃. Major disease categories are based on the chapter division of the ICD-10 diagnostic coding system. Single-day exposure on the same day (lag 0) was used as the exposure metric of PM₂.₅. Two-day moving average exposure (lag 0–1) was used as the exposure metric of O₃. In single-pollutant models, the effects of PM₂.₅ and O₃ were estimated without adjustment for co-pollutants; in 2-pollutant models, the effects of PM₂.₅ were estimated after adjustment for O₃, and the effects of O₃ were estimated after adjustment for PM₂.₅. The Benjamini–Hochberg procedure was applied to adjust the $P$ values across the 14 major disease categories. *Statistically significant estimate ($P < 0.05$).

**Table 3. Percent change in hospital admissions per 10-μg/m³ increase in PM$_{2.5}$ and O$_3$ by minor disease category, on average across all cities.**

| CCS code | Minor disease category | Percent change in admissions per 10-μg/m³ increase in PM$_{2.5}$ | | | | Percent change in admissions per 10-μg/m³ increase in O$_3$ | | | |
| --- | --- | --- | --- | --- | --- | --- | --- | --- | --- |
| | | Single-pollutant model | | Two-pollutant model | | Single-pollutant model | | Two-pollutant model | |
| | | Point estimate (95% CI) | Adjusted $P$ value | Point estimate (95% CI) | Adjusted $P$ value | Point estimate (95% CI) | Adjusted $P$ value | Point estimate (95% CI) | Adjusted $P$ value |
| 3 | Bacterial infection, unspecified site | 0.87 (0.44, 1.29) | <0.001* | 0.93 (0.40, 1.46) | 0.006* | −0.06 (−0.59, 0.46) | 0.919 | −0.22 (−0.81, 0.38) | 0.877 |
| 13 | Cancer of stomach | 0.18 (0.03, 0.33) | 0.077 | 0.24 (0.07, 0.42) | 0.038* | 0.09 (−0.12, 0.30) | 0.842 | 0.06 (−0.16, 0.29) | 0.920 |
| 19 | Cancer of bronchus, lung | 0.27 (0.15, 0.40) | <0.001* | 0.29 (0.16, 0.42) | <0.001* | 0.07 (−0.07, 0.20) | 0.784 | 0.07 (−0.07, 0.22) | 0.814 |
| 24 | Cancer of breast | 0.34 (0.19, 0.48) | <0.001* | 0.37 (0.19, 0.54) | <0.001* | 0.08 (−0.10, 0.26) | 0.839 | 0.05 (−0.14, 0.24) | 0.920 |
| 45 | Maintenance chemotherapy; radiotherapy | 0.10 (0.02, 0.18) | 0.068 | 0.11 (0.03, 0.18) | 0.030* | −0.06 (−0.14, 0.02) | 0.626 | −0.07 (−0.15, 0.02) | 0.707 |
| 49 | Diabetes mellitus without complication | 0.23 (0.11, 0.35) | 0.002* | 0.27 (0.13, 0.41) | 0.001* | 0.10 (−0.04, 0.25) | 0.674 | 0.04 (−0.12, 0.20) | 0.931 |
| 50 | Diabetes mellitus with complications | 0.30 (0.18, 0.41) | <0.001* | 0.35 (0.21, 0.49) | <0.001* | 0.20 (0.04, 0.35) | 0.164 | 0.10 (−0.07, 0.27) | 0.808 |
| 54 | Gout and other crystal arthropathies | 0.53 (0.16, 0.90) | 0.027* | 0.71 (0.23, 1.18) | 0.024* | 0.06 (−0.33, 0.44) | 0.919 | −0.06 (−0.48, 0.36) | 0.943 |
| 59 | Deficiency and other anemia | 0.29 (0.11, 0.47) | 0.012* | 0.37 (0.16, 0.59) | 0.006* | −0.05 (−0.25, 0.16) | 0.919 | −0.09 (−0.31, 0.13) | 0.814 |
| 72 | Anxiety, somatoform, dissociative, and personality disorders | 0.29 (0.07, 0.52) | 0.049* | 0.31 (0.05, 0.58) | 0.080 | 0.23 (−0.02, 0.47) | 0.554 | 0.21 (−0.06, 0.47) | 0.707 |
| 79 | Parkinson disease | 0.50 (0.12, 0.88) | 0.047* | 0.57 (0.14, 1.00) | 0.049* | −0.02 (−0.46, 0.42) | 0.963 | −0.14 (−0.56, 0.28) | 0.890 |
| 93 | Conditions associated with dizziness or vertigo | 0.34 (0.15, 0.53) | 0.003* | 0.27 (0.05, 0.49) | 0.067 | −0.00 (−0.22, 0.22) | 0.992 | −0.08 (−0.32, 0.15) | 0.877 |
| 97 | Peri-, endo-, and myocarditis; cardiomyopathy (except that caused by tuberculosis or sexually transmitted disease) | 0.28 (0.08, 0.48) | 0.036* | 0.30 (0.06, 0.54) | 0.068 | 0.09 (−0.14, 0.32) | 0.888 | 0.03 (−0.22, 0.27) | 0.958 |
| 98 | Essential hypertension | 0.22 (0.10, 0.34) | 0.003* | 0.29 (0.14, 0.44) | 0.001* | −0.18 (−0.31, −0.05) | 0.114 | −0.24 (−0.38, −0.10) | 0.035* |
| 100 | Acute myocardial infarction | 0.38 (0.24, 0.52) | <0.001* | 0.35 (0.18, 0.52) | <0.001* | 0.20 (0.03, 0.38) | 0.243 | 0.15 (−0.04, 0.34) | 0.707 |
| 101 | Coronary atherosclerosis and other heart disease | 0.30 (0.22, 0.37) | <0.001* | 0.29 (0.21, 0.37) | <0.001* | 0.14 (0.04, 0.24) | 0.102 | 0.06 (−0.04, 0.17) | 0.790 |
| 103 | Pulmonary heart disease | 0.58 (0.27, 0.89) | 0.002* | 0.53 (0.17, 0.89) | 0.024* | 0.31 (−0.09, 0.70) | 0.636 | 0.18 (−0.23, 0.59) | 0.814 |
| 106 | Cardiac dysrhythmias | 0.30 (0.16, 0.44) | <0.001* | 0.28 (0.13, 0.43) | 0.002* | 0.21 (0.06, 0.36) | 0.114 | 0.21 (0.05, 0.38) | 0.277 |
| 108 | Congestive heart failure, nonhypertensive | 0.45 (0.24, 0.67) | <0.001* | 0.42 (0.18, 0.67) | 0.007* | 0.06 (−0.20, 0.32) | 0.919 | −0.11 (−0.39, 0.16) | 0.814 |
| 109 | Acute cerebrovascular disease | 0.16 (0.09, 0.23) | <0.001* | 0.20 (0.12, 0.28) | <0.001* | −0.02 (−0.11, 0.08) | 0.919 | −0.05 (−0.15, 0.05) | 0.814 |
| 111 | Other and ill-defined cerebrovascular disease | 0.24 (0.08, 0.39) | 0.017* | 0.30 (0.12, 0.48) | 0.008* | −0.13 (−0.31, 0.05) | 0.681 | −0.13 (−0.32, 0.07) | 0.759 |
| 112 | Transient cerebral ischemia | 0.26 (0.14, 0.38) | <0.001* | 0.33 (0.19, 0.47) | <0.001* | −0.00 (−0.15, 0.14) | 0.986 | −0.07 (−0.22, 0.09) | 0.814 |
| 113 | Late effects of cerebrovascular disease | 0.32 (0.08, 0.57) | 0.049* | 0.33 (0.03, 0.63) | 0.098 | 0.06 (−0.21, 0.32) | 0.919 | −0.04 (−0.32, 0.25) | 0.943 |

(*Continued*)

**Table 3.** (Continued)

| CCS code | Minor disease category | Percent change in admissions per 10-µg/m³ increase in PM$_{2.5}$ | | | | Percent change in admissions per 10-µg/m³ increase in O$_3$ | | | |
|---|---|---|---|---|---|---|---|---|---|
| | | Single-pollutant model | | Two-pollutant model | | Single-pollutant model | | Two-pollutant model | |
| | | Point estimate (95% CI) | Adjusted P value | Point estimate (95% CI) | Adjusted P value | Point estimate (95% CI) | Adjusted P value | Point estimate (95% CI) | Adjusted P value |
| 122 | Pneumonia (except that caused by tuberculosis or sexually transmitted disease) | 0.36 (0.28, 0.44) | <0.001* | 0.28 (0.19, 0.37) | <0.001* | 0.29 (0.19, 0.39) | <0.001* | 0.21 (0.10, 0.32) | 0.007* |
| 125 | Acute bronchitis | 0.32 (0.18, 0.46) | <0.001* | 0.32 (0.16, 0.48) | 0.001* | 0.30 (0.14, 0.46) | 0.010* | 0.19 (0.02, 0.36) | 0.370 |
| 126 | Other upper respiratory infections | 0.31 (0.17, 0.44) | <0.001* | 0.28 (0.13, 0.44) | 0.004* | 0.43 (0.28, 0.58) | <0.001* | 0.44 (0.28, 0.59) | <0.001* |
| 127 | Chronic obstructive pulmonary disease and bronchiectasis | 0.52 (0.40, 0.64) | <0.001* | 0.51 (0.38, 0.64) | <0.001* | 0.28 (0.14, 0.42) | 0.002* | 0.10 (−0.04, 0.24) | 0.718 |
| 128 | Asthma | −0.11 (−0.39, 0.16) | 0.630 | −0.39 (−0.71, −0.07) | 0.073 | 0.72 (0.39, 1.04) | <0.001* | 0.75 (0.43, 1.08) | <0.001* |
| 133 | Other lower respiratory disease | 0.28 (0.17, 0.39) | <0.001* | 0.30 (0.17, 0.43) | <0.001* | 0.06 (−0.07, 0.19) | 0.799 | −0.06 (−0.19, 0.08) | 0.816 |
| 134 | Other upper respiratory disease | 0.31 (0.17, 0.45) | <0.001* | 0.25 (0.09, 0.42) | 0.019* | 0.29 (0.13, 0.45) | 0.014* | 0.21 (0.04, 0.39) | 0.310 |
| 135 | Intestinal infection | 0.76 (0.40, 1.12) | <0.001* | 0.97 (0.59, 1.35) | <0.001* | 0.10 (−0.23, 0.44) | 0.919 | −0.22 (−0.59, 0.14) | 0.790 |
| 138 | Esophageal disorders | 0.34 (0.10, 0.59) | 0.031* | 0.39 (0.11, 0.67) | 0.038* | 0.04 (−0.31, 0.39) | 0.919 | −0.11 (−0.48, 0.26) | 0.904 |
| 140 | Gastritis and duodenitis | 0.24 (0.08, 0.40) | 0.018* | 0.24 (0.06, 0.42) | 0.049* | 0.00 (−0.16, 0.16) | 0.990 | −0.07 (−0.26, 0.12) | 0.877 |
| 142 | Appendicitis and other appendiceal conditions | 0.26 (0.12, 0.41) | 0.003* | 0.25 (0.08, 0.42) | 0.024* | 0.02 (−0.14, 0.19) | 0.919 | −0.03 (−0.20, 0.15) | 0.943 |
| 148 | Peritonitis and intestinal abscess | 0.57 (0.13, 1.00) | 0.049* | 0.40 (−0.10, 0.91) | 0.293 | 0.29 (−0.24, 0.82) | 0.759 | 0.20 (−0.35, 0.76) | 0.877 |
| 149 | Biliary tract disease | 0.07 (−0.03, 0.17) | 0.348 | 0.16 (0.04, 0.27) | 0.033* | −0.03 (−0.16, 0.10) | 0.919 | −0.03 (−0.16, 0.11) | 0.943 |
| 151 | Other liver diseases | 0.25 (0.12, 0.38) | 0.001* | 0.22 (0.08, 0.37) | 0.019* | 0.22 (0.07, 0.36) | 0.077 | 0.14 (−0.01, 0.30) | 0.630 |
| 153 | Gastrointestinal hemorrhage | 0.30 (0.14, 0.46) | 0.002* | 0.34 (0.14, 0.53) | 0.007* | 0.11 (−0.08, 0.29) | 0.745 | 0.00 (−0.21, 0.21) | 1.000 |
| 154 | Noninfectious gastroenteritis | 0.44 (0.23, 0.64) | <0.001* | 0.43 (0.21, 0.64) | 0.001* | 0.29 (0.08, 0.50) | 0.114 | 0.18 (−0.06, 0.42) | 0.707 |
| 156 | Nephritis, nephrosis, and renal sclerosis | 0.23 (0.08, 0.39) | 0.020* | 0.28 (0.09, 0.46) | 0.024* | 0.16 (−0.02, 0.34) | 0.568 | 0.16 (−0.04, 0.35) | 0.707 |
| 158 | Chronic renal failure | 0.32 (0.19, 0.45) | <0.001* | 0.32 (0.17, 0.47) | <0.001* | −0.02 (−0.18, 0.14) | 0.919 | −0.10 (−0.26, 0.07) | 0.808 |
| 160 | Calculus of urinary tract | 0.29 (0.13, 0.45) | 0.003* | 0.30 (0.10, 0.49) | 0.019* | 0.21 (0.03, 0.38) | 0.207 | 0.17 (−0.02, 0.35) | 0.646 |
| 161 | Other diseases of kidney and ureters | 0.22 (0.06, 0.38) | 0.035* | 0.19 (−0.00, 0.37) | 0.171 | 0.25 (0.08, 0.42) | 0.102 | 0.23 (0.04, 0.41) | 0.357 |
| 199 | Chronic ulcer of skin | 0.91 (0.33, 1.48) | 0.013* | 0.99 (0.31, 1.68) | 0.025* | 0.32 (−0.33, 0.97) | 0.784 | 0.02 (−0.66, 0.70) | 0.996 |
| 205 | Spondylosis, intervertebral disc disorders, and other back problems | 0.27 (0.14, 0.40) | <0.001* | 0.33 (0.19, 0.47) | <0.001* | 0.07 (−0.06, 0.19) | 0.749 | 0.03 (−0.11, 0.17) | 0.943 |

(*Continued*)

**Table 3.** (Continued)

| CCS code | Minor disease category | Percent change in admissions per 10-μg/m$^3$ increase in PM$_{2.5}$ | | | | Percent change in admissions per 10-μg/m$^3$ increase in O$_3$ | | | |
|---|---|---|---|---|---|---|---|---|---|
| | | Single-pollutant model | | Two-pollutant model | | Single-pollutant model | | Two-pollutant model | |
| | | Point estimate (95% CI) | Adjusted *P* value | Point estimate (95% CI) | Adjusted *P* value | Point estimate (95% CI) | Adjusted *P* value | Point estimate (95% CI) | Adjusted *P* value |
| 212 | Other bone disease and musculoskeletal deformities | 0.27 (−0.05, 0.58) | 0.238 | 0.51 (0.14, 0.89) | 0.038* | −0.15 (−0.54, 0.25) | 0.890 | −0.18 (−0.59, 0.24) | 0.814 |

Results are presented as point estimates and 95% CIs of the percentage increase in daily hospital admissions associated with a 10-μg/m$^3$ increase in PM$_{2.5}$ or O$_3$. Minor disease categories are based on the Clinical Classifications Software (CCS). Single-day exposure on the same day (lag 0) was used as the exposure metric of PM$_{2.5}$. Two-day moving average exposure (lag 0–1) was used as the exposure metric of O$_3$. In single-pollutant models, the effects of PM$_{2.5}$ and O$_3$ were estimated without adjustment for co-pollutants; in 2-pollutant models, the effects of PM$_{2.5}$ were estimated after adjustment for O$_3$, and the effects of O$_3$ were estimated after adjustment for PM$_{2.5}$. The Benjamini–Hochberg procedure was applied to adjust the *P* values across the 188 minor disease categories. The 46 minor disease categories significantly associated with PM$_{2.5}$ or O$_3$ in single- or 2-pollutant models after multiple comparisons adjustment are listed here; results for all disease categories are shown in S6–S19 Tables.
*Statistically significant estimate ($P < 0.05$).

infection, esophageal disorders, gastritis and duodenitis, appendiceal conditions, liver diseases, gastrointestinal hemorrhage, and noninfectious gastroenteritis; 3 categories of genitourinary diseases, including chronic renal failure, calculus of urinary tract, and nephritis, nephrosis, and renal sclerosis; and 10 categories of other diseases, including bacterial infection, cancer of bronchus and lung, cancer of breast, diabetes mellitus without/with complications, crystal arthropathies, anemia, Parkinson disease, chronic ulcer of skin, and spondylosis, intervertebral disc disorders, and other back problems. For example, each 10-μg/m$^3$ increase in same-day PM$_{2.5}$ was associated with a 0.34% (95% CI 0.14% to 0.53%; adjusted $P = 0.007$) increase in hospital admissions for gastrointestinal hemorrhage when adjusted for O$_3$. The 2-day moving average concentration of O$_3$ (lag 0–1 days) was significantly positively associated with hospital admissions for pneumonia, upper respiratory infections, and asthma in both single- and 2-pollutant models, as well as acute bronchitis, COPD and bronchiectasis, and other upper respiratory disease in single-pollutant models only.

The results of the subgroup analyses are shown in Fig 2, S6 and S7 Figs. We found stronger associations between PM$_{2.5}$ and most disease categories in people aged 65–74 years or ≥75 years than in people aged <65 years; for several disease categories, the effect modification was statistically significant (in an exploratory analysis without adjustment for multiple comparisons) (Fig 2 and S6 Fig). However, we did not observe the same pattern for O$_3$ (S7 Fig). The evidence of effect modification by sex, season, and region (using both 2- and 6-region divisions) was mixed and uncertain (S6, S7, S12 and S13 Figs). The results of the exposure–response analyses are shown in Fig 3, S8 and S9 Figs. The exposure–response curves between PM$_{2.5}$ and hospital admissions generally showed steeper slopes at lower concentrations than at higher concentrations, with no safety threshold below which PM$_{2.5}$ exposure had no health effect (Fig 3 and S8 Fig). A threshold of O$_3$ in the range of 40 to 50 μg/m$^3$ (near the background level of O$_3$) was observed in the exposure–response curve between O$_3$ and hospital admissions for diseases of the respiratory system (S9 Fig).

The results of the sensitivity analyses are shown in S10 and S11 Figs. Our results remained stable using alternative *df* for the smoothing functions of temperature and relative humidity. The results for PM$_{2.5}$ remained stable using alternative *df* for the smoothing function of calendar day. The results for O$_3$ were substantially biased when using too few *df* for the smoothing function of calendar day (e.g., the association between O$_3$ and hospital admissions for diseases

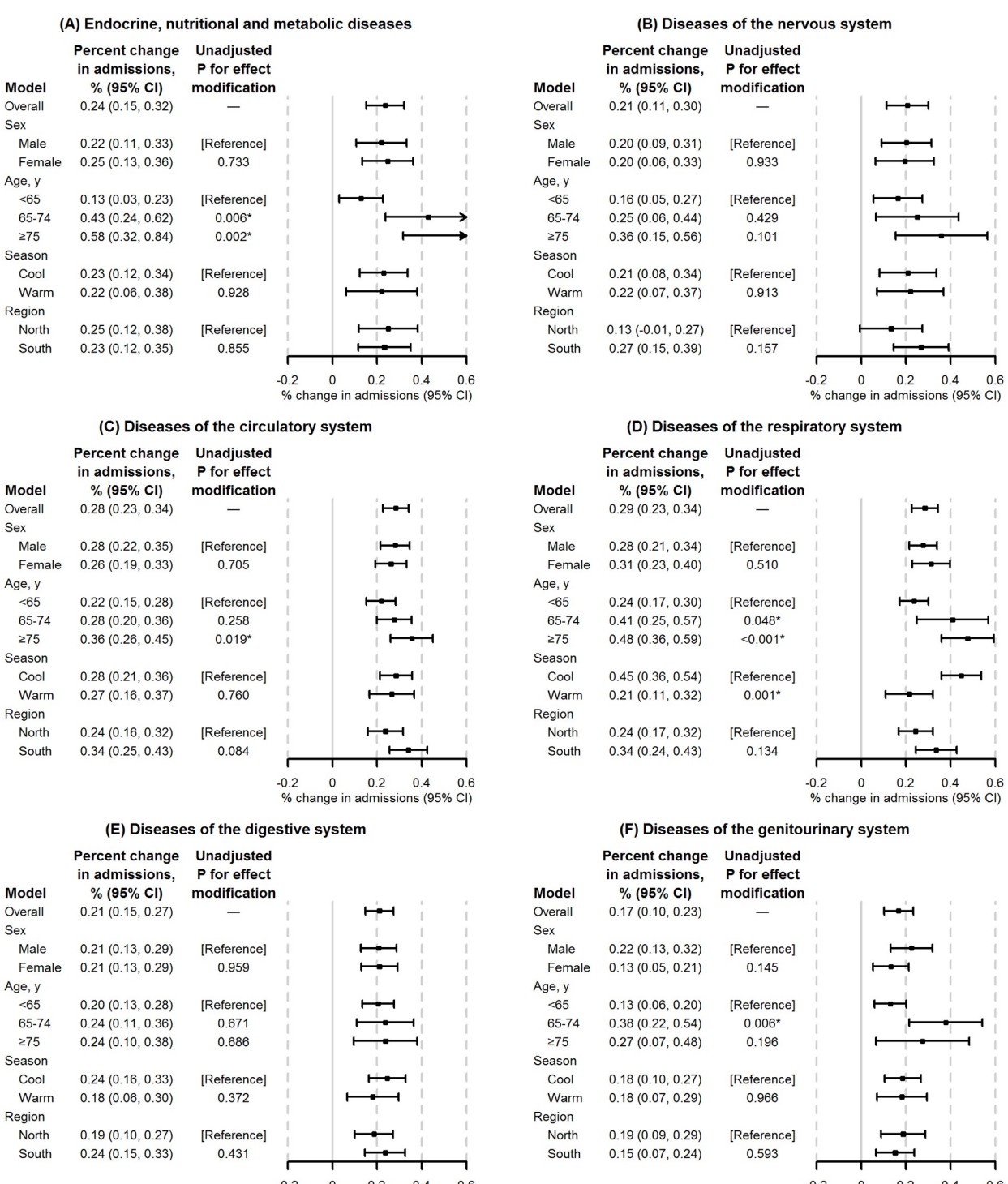

**Fig 2. Percent change in hospital admissions per 10-μg/m³ increase in PM₂.₅ by major disease category for study subgroups, on average across all cities.** (A) Endocrine, nutritional, and metabolic diseases; (B) nervous diseases; (C) circulatory diseases; (D) respiratory diseases; (E) digestive diseases; (F) genitourinary diseases. Results are presented as point estimates and 95% CIs of the percentage increase in daily hospital admissions associated with a 10-μg/m³ increase in PM₂.₅. Major disease categories are based on the chapter division of the ICD-10 diagnostic coding system. Single-day exposure on the same day (lag 0) was used as the exposure metric of PM₂.₅. The effects of PM₂.₅ were estimated after adjustment for O₃. The cool season is from October to March; the warm season is from April to September. The 2 regions of China (North and South) are divided by the Huai River–Qinling Mountain line. The *P* values were not adjusted for multiple comparisons. Results for all disease categories and for O₃ are shown in S6 and S7 Figs. *Statistically significant estimate (*P* < 0.05).

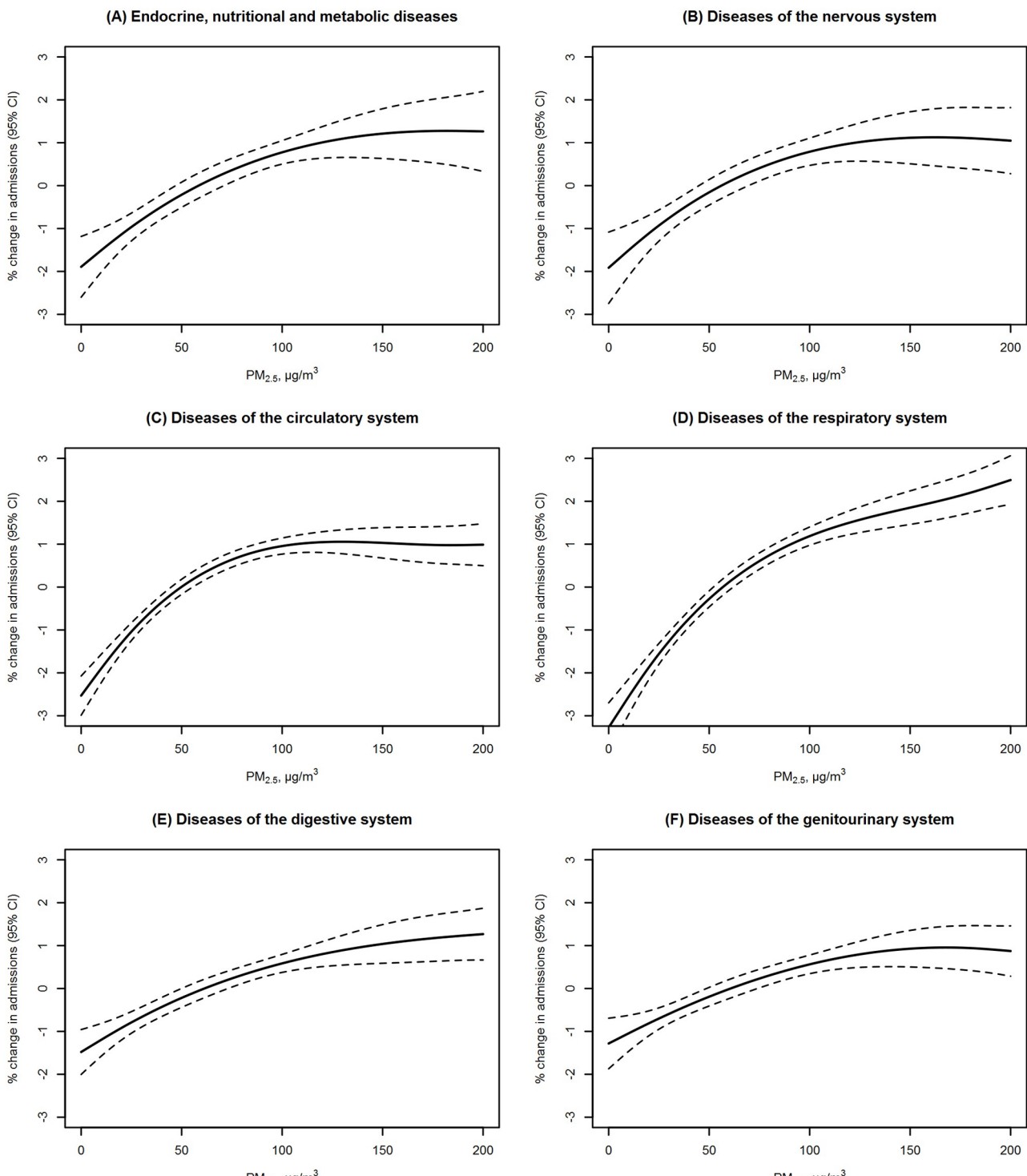

**Fig 3. Exposure–response curves of the effects of PM₂.₅ on hospital admissions by major disease category, on average across all cities.** (A) Endocrine, nutritional, and metabolic diseases; (B) nervous diseases; (C) circulatory diseases; (D) respiratory diseases; (E) digestive diseases; (F) genitourinary diseases. The vertical scale represents the relative change from the mean effect of PM₂.₅ on daily hospital admissions, with solid lines indicating point estimates and dashed lines indicating 95% CIs. Major disease categories are based on the chapter division of the ICD-10 diagnostic coding system. Single-day exposure on the same day (lag 0) was used as the exposure metric of PM₂.₅. The national average exposure–response curves were developed using multivariate meta-analysis approaches. Results for all disease categories and for O₃ are shown in S8 and S9 Figs.

**Table 4. Annual reduction in hospital admissions and hospitalization expenses attributable to a 10-μg/m$^3$ reduction in daily PM$_{2.5}$ level in China.**

| Major disease category | Annual reduction in hospital admissions (95% CI) | Annual reduction in hospitalization expenses, million ¥ (95% CI) |
|---|---|---|
| All | 120,145 (107,698 to 132,592) | 1,011.9 (903.4 to 1,120.4) |
| Certain infectious and parasitic diseases | 3,616 (746 to 6,487) | 20.0 (4.1 to 35.8) |
| Neoplasms | 6,948 (3,961 to 9,936) | 107.2 (61.1 to 153.3) |
| Diseases of the blood and blood-forming organs and certain disorders involving the immune mechanism | 1,230 (369 to 2,090) | 8.3 (2.5 to 14.1) |
| Endocrine, nutritional, and metabolic diseases | 5,405 (3,467 to 7,343) | 43.3 (27.8 to 58.8) |
| Mental and behavioral disorders | 950 (257 to 1,644) | 6.6 (1.8 to 11.5) |
| Diseases of the nervous system | 4,727 (2,581 to 6,872) | 35.3 (19.3 to 51.3) |
| Diseases of the eye and adnexa | 1,832 (-817 to 4,480) | 10.9 (-4.9 to 26.7) |
| Diseases of the ear and mastoid process | 1,242 (227 to 2,256) | 7.1 (1.3 to 12.9) |
| Diseases of the circulatory system | 32,968 (26,343 to 39,594) | 337.0 (269.3 to 404.7) |
| Diseases of the respiratory system | 30,030 (23,855 to 36,205) | 170.2 (135.2 to 205.1) |
| Diseases of the digestive system | 16,110 (11,297 to 20,924) | 131.8 (92.4 to 171.2) |
| Diseases of the skin and subcutaneous tissue | 1,412 (459 to 2,364) | 8.4 (2.7 to 14.0) |
| Diseases of the musculoskeletal system and connective tissue | 6,109 (3,791 to 8,428) | 64.0 (39.7 to 88.2) |
| Diseases of the genitourinary system | 7,566 (4,625 to 10,508) | 61.8 (37.8 to 85.8) |

The estimation was based on the total number of hospital admissions and the average cost for each hospitalization in China in 2016, collected from China Health and Family Planning Statistical Yearbook 2017. ¥ = Chinese yuan.

of the respiratory system even became negative when $df$ = 6 per year), indicating that the confounding effects of seasonality and/or long-term patterns were not controlled adequately in that case; however, the bias disappeared with $df \geq 8$ per year, and the results remained stable with increasing $df$ up to 14 per year, demonstrating that the models used in the main analyses had successfully removed the confounding effects. None of the 8 negative control outcomes were associated with PM$_{2.5}$ or O$_3$ in single- or 2-pollutant models, which provided further reassurance that our findings were not substantially confounded or biased (S20 Table).

Table 4 provides the estimated annual reduction in cause-specific hospital admissions and hospitalization expenses attributable to a 10-μg/m$^3$ reduction in daily PM$_{2.5}$ level in China. Based on hospital admissions and costs in 2016, a 10-μg/m$^3$ reduction in PM$_{2.5}$ would reduce the number of hospital admissions by 120,145 (95% CI 107,698 to 132,592) and hospitalization expenses by 1,011.9 (95% CI 903.4 to 1,120.4) million yuan nationwide, of which 52.4% of reduced cases and 50.1% of reduced costs would be attributable to reductions in diseases of the circulatory and respiratory systems, with reductions in non-cardiorespiratory diseases accounting for the other 47.6% of reduced cases and 49.9% of reduced costs.

## Discussion

This national study systematically assessed the acute effects of main air pollutants on hospital admissions for 14 major and 188 minor disease categories using standardized analytic approaches, providing a unified perspective on the health effects of air pollution on multiple organ systems, with minimal publication bias. We found robust positive associations between PM$_{2.5}$ and 7 major disease categories and between O$_3$ and respiratory diseases, as well as significant evidence that diseases in dozens of minor disease categories—many are non-cardiorespiratory diseases—can be triggered and/or exacerbated by short-term exposure to air pollution. Consistent with prior research [4,20,31,36,37], we generally observed higher risks related to

$PM_{2.5}$ in older people (65–74 or $\geq$75 years) than in younger people (<65 years), and steeper slopes of exposure–response curves between $PM_{2.5}$ and hospital admissions at lower concentrations than at higher concentrations.

Our results regarding cardiorespiratory diseases are in line with the current understanding of the deleterious impacts of air pollution on the circulatory and respiratory systems. The well-known outcomes associated with air pollutants serve as positive controls for this study. The size of effects in this analysis is generally similar to those of recent multisite studies in China [20,31,36,37], and smaller than those from North America and Europe [1,2,4]. The lower effect estimates of this study than those from high-income countries may have at least 2 possible explanations. First, the levels of particulate air pollution were much higher in China than in the high-income countries. As observed in our study and previous multisite studies [4,20,31,37], the slopes of exposure–response relationships were generally steeper at lower concentrations than at higher concentrations, which may be partly attributable to larger exposure measurement errors on heavily polluted days, when public health interventions would be reinforced and people would more actively take self-protection measures. Second, the chemical profile of $PM_{2.5}$ was different between China and the high-income countries. $PM_{2.5}$ in China's air had a higher proportion of crustal materials, which may have lower toxicity than components that originate from fossil fuel combustion [31,37]. We demonstrated that ambient exposure to $PM_{2.5}$ was robustly associated with hospital admissions for endocrine, nutritional, and metabolic diseases and diseases of the nervous system in the Chinese population. We also identified diabetes mellitus without/with complications and Parkinson disease as typical minor disease categories that were significantly associated with same-day $PM_{2.5}$ in both single- and 2-pollutant models. These results support the growing evidence that acute exposure to air pollution can increase the risk of certain diseases of these organ systems [45–47].

More importantly, by inspecting the associations of air pollution with a broad spectrum of diseases, this study provides new insights into the health effects of air pollution that were scarcely or never reported before. We found a robust positive association between $PM_{2.5}$ and hospitalizations for diseases of the digestive system. Previous studies have suggested that air pollution exposure is associated with the risk of some gastrointestinal diseases, e.g., gastroenteric disorders in young children [48], acute appendicitis [49], inflammatory bowel diseases [50], and peptic ulcer bleeding [10]. For example, Tian et al. [10] reported a positive association of daily air pollution and emergency admissions for peptic ulcer bleeding using the data of 8,566 recorded cases in Hong Kong's elderly population during 2005–2010, which was one of the largest studies to date on this topic. Exposure to $PM_{2.5}$ was found to induce a nonalcoholic steatohepatitis-like phenotype and hepatic fibrosis in animal models [51,52]; evidence from human data is limited. In this analysis, we observed consistent associations between same-day $PM_{2.5}$ and various digestive diseases, including intestinal infection, esophageal disorders, gastritis and duodenitis, appendiceal conditions, liver diseases ($n$ = 1,398,024), gastrointestinal hemorrhage ($n$ = 811,588), and noninfectious gastroenteritis. Our findings strengthen the hypothesis that air pollution exposure adversely affects certain diseases of the digestive system, which should be fully considered in policy making aimed at protecting public health from air pollution.

We also found a robust positive association between $PM_{2.5}$ and hospitalizations for diseases of the genitourinary system. A Chinese biopsy series study found that long-term exposure to $PM_{2.5}$ was associated with the odds for membranous nephropathy [53]. Recent cohort studies in US veterans [12,13] and Taiwanese residents [14] reported that elevated levels of $PM_{2.5}$ and other pollutants were associated with increased risk of incident chronic kidney disease, chronic kidney disease progression, and end-stage renal disease. In this analysis, we observed consistent associations between same-day $PM_{2.5}$ and several genitourinary diseases, including

nephritis, nephrosis, and renal sclerosis; chronic renal failure; and calculus of urinary tract. Our findings indicate that even being exposed to air pollution for a short time can have adverse effects on one's genitourinary system, highlighting the importance of prompt public health intervention, especially for susceptible populations.

In addition to the above, we found a robust positive association between $PM_{2.5}$ and hospitalizations for diseases of the musculoskeletal system and connective tissue. Also noteworthy is that higher $PM_{2.5}$ exposure was significantly associated with increased risk of hospitalization for crystal arthropathies, anemia, chronic ulcer of skin, and spondylosis, intervertebral disc disorders, and other back problems in both single- and 2-pollutant models, as well as anxiety, somatoform, dissociative, and personality disorders and conditions associated with dizziness or vertigo in single-pollutant models. Chronic ulcer of skin, which had a relatively small sample size ($n$ = 61,605) among all minor disease categories, had one of the largest effect estimates of $PM_{2.5}$ among all minor disease categories. Overall, these findings are novel—some associations are supported by previous studies, e.g., $PM_{2.5}$ and anxiety disorders [54–56], while others are reported to our knowledge for the first time. Certain musculoskeletal diseases and mental disorders that were associated with $PM_{2.5}$ in this study contribute to the global burden of disease especially in terms of nonfatal consequences. For example, low back pain has prevailed as the top cause of years lived with disability worldwide for nearly 3 decades [57,58]. Our findings indicate that air pollution may be a novel risk factor for the diseases that were associated with particulate matter in this study, and has the potential to account for the unattributed disease burden [59].

We estimated a substantial annual reduction in the number of hospital admissions and hospitalization expenses that would be attributable to an improvement in daily $PM_{2.5}$ in China, indicating that the government should develop effective mitigation policies to reduce the burden of disease attributable to air pollution. According to the cause-specific estimates, 47.6% of reduced hospital admissions and 49.9% of reduced hospitalization expenses would be attributable to non-cardiorespiratory diseases. In contrast, a time-series study using mortality data in 272 Chinese cities reported that only 15.2% of premature deaths prevented by $PM_{2.5}$ reduction would be attributable to non-cardiorespiratory diseases [31]. It should be noted that our estimates are based on short-term associations between air pollution and daily hospital admissions, and thus are not comparable to estimates based on long-term associations, e.g., in the Global Burden of Disease Study [5]; however, our estimates can be compared with those from studies of the same design [31]. Therefore, the precise estimation of the morbidity burden attributable to air pollution in China and worldwide deserves further research.

The biological mechanisms underlying the associations between air pollution and cardiorespiratory diseases are relatively well understood. Epidemiological and toxicological studies have demonstrated that exposure to particulate air pollution may induce systemic inflammation and oxidative stress, cause autonomic imbalance favoring sympathetic tone, and activate the hypothalamic–pituitary–adrenal axis, leading to vasoconstriction, hypertension, tachycardia, reduced heart rate variability, increased plasma viscosity, dyslipidemia, vascular endothelial dysfunction, atherosclerosis progression, platelet aggregation, hypercoagulability, and increased thrombogenesis [6,7,15]. One or more of these mechanistic pathways may be involved in the extrapulmonary effects of air pollution [60,61]. Moreover, there are a series of hypotheses to explain the associations between air pollution and specific non-cardiorespiratory diseases, some of which have been supported by emerging clinical and experimental evidence. Short- and long-term exposures to $PM_{2.5}$ and other air pollutants have been suggested to adversely affect glucose and insulin homeostasis, leading to glucose intolerance, decreased insulin sensitivity, increased serum lipid levels, and, finally, a higher risk of diabetes mellitus as well as other metabolic diseases [16,45,62]. Air pollution exposure has been reported to be

associated with several key factors in the pathophysiology of central nervous system diseases, including neuro-inflammation and oxidative stress, microglial activation, dopaminergic neuron damage, and cerebrovascular impairment [46,47]. These pathways may also play an important role in the pathogenesis of mental and behavioral disorders [54–56]. Air pollutants can be transported to the gastrointestinal tract by either mucociliary clearance from the lungs or ingestion of polluted food and water, and may exert direct toxic effects on gastrointestinal epithelial cells and influence gut microbial composition, leading to increased intestinal permeability, altered intestinal immunity, and oxidative stress and inflammatory response [10,63,64]. The skin, the largest and outermost organ of the body, may be adversely affected by pollutants via direct exposure from the air or indirect exposure from the systemic circulation [65]. The kidney, a highly vascularized organ, is vulnerable to both macro- and microvascular dysfunction and thus is likely to be affected through the pathways of air-pollution-induced cardiovascular toxicity [12,66]. Finally, we postulate that air pollution exposure may increase the pain of various conditions (e.g., renal colic caused by kidney stones and chronic back pain) through prostaglandin pathways. Prostaglandins are major proinflammatory mediators that can sensitize pain receptors [67,68]. Increased levels of prostaglandins cause more pain. Recent animal and human studies have linked excessive prostaglandins with short- and long-term exposures to $PM_{2.5}$ and other air pollutants [67–70]. Nevertheless, the exact biological mechanisms underlying the reported associations remain elusive and deserve further research.

This study has some strengths. First, by analyzing all causes of hospitalization rather than a few prespecified outcomes, and reporting all results, this study provides evidence for health effects of air pollution that may have been neglected due to lack of prior knowledge. Second, this study defined disease categories at 2 levels. The major disease categories were used to evaluate the overall health effects of air pollution on each organ system, while the minor disease categories were used to identify the specific diseases that can be triggered and/or exacerbated by air pollution exposure. Third, this study has the advantages of a large dataset with good internal consistency in data collection, good timeliness (2013–2017), and extensive geographical and population coverage (252 cities in China; all ages), ensuring the generalizability of our results to the Chinese population or other populations in similar settings. Fourth, this study also provides the exposure–response relationships between air pollution and cause-specific hospital admissions, which support the hypothesis that there is no safety threshold for the health effects of $PM_{2.5}$ exposure, but there may be a safety threshold close to the background concentration of $O_3$ for the effect of $O_3$ exposure on respiratory diseases. The possible existence of a threshold for $O_3$ has been suggested by several studies conducted in high-income countries [4,43], and should be further investigated in low- and middle-income countries, where the evidence is still limited.

This study has some limitations. First, we used monitored ambient concentrations of air pollutants as the proxy for personal exposures, leading to exposure measurement errors that are expected to bias the effect estimates toward null [71]. Second, we cannot rule out the possibility of misclassifications resulting from diagnostic or coding errors in this large-scale nationwide dataset. Generally, the more specific the diagnostic categories that are used, the greater the likelihood of outcome misclassification. The diagnosis coding accuracy for the minor disease categories may thus be lower than that for the major disease categories. Therefore, our results concerning minor disease categories should be interpreted with caution. Future studies based on medically reviewed outcome data are warranted to confirm our findings. Third, our data cannot distinguish between scheduled and unscheduled hospitalizations. Including scheduled hospital admissions (e.g., planned surgeries), which are inherently unrelated to air pollution, is expected to reduce the estimation precision. However, a general-practitioner-based referral system is not available in China [19,20]. Hospital visits and admissions are

generally unscheduled and are on a first-come, first-served basis. Therefore, the impact of scheduled hospitalizations on our results is expected to be minor. Fourth, we did not have data on the chemical composition or emission sources of particulate matter, hindering us from further research. Fifth, we cannot estimate the long-term risk related to air pollution under the time-series analysis design. Studies of other designs (e.g., cohort studies) should be conducted to entirely understand the hazardous effects of air pollution, as well as replications of this work in other populations to confirm the external validity of our results.

## Conclusions

This study provides a comprehensive picture of the associations between short-term exposure to main air pollutants and cause-specific risk of hospital admission in China over a wide spectrum of human diseases. Our analysis showed that air pollution exposure was associated with increased risk of hospitalization for diseases of multiple organ systems, including certain diseases of the digestive, musculoskeletal, and genitourinary systems; many of these associations are important but still not fully recognized. This study also evaluated potential effect modifiers and exposure–response relationships between air pollution and hospitalization. These results can inform policy making aimed at protecting public health from air pollution in China.

## Supporting information

**S1 Fig. Locations of the 252 Chinese cities in this study.** The base map was created using the open-source R package *maps*.
(TIF)

**S2 Fig. Percent change in hospital admissions per 10-μg/m$^3$ increase in PM$_{2.5}$ by major disease category using different lag structures, on average across all cities.**
(DOCX)

**S3 Fig. Percent change in hospital admissions per 10-μg/m$^3$ increase in O$_3$ by major disease category using different lag structures, on average across all cities.**
(DOCX)

**S4 Fig. Percent change in hospital admissions per 10-μg/m$^3$ increase in PM$_{2.5}$ by major disease category using 3-pollutant models, on average across all cities.**
(DOCX)

**S5 Fig. Percent change in hospital admissions per 10-μg/m$^3$ increase in O$_3$ by major disease category using 3-pollutant models, on average across all cities.**
(DOCX)

**S6 Fig. Percent change in hospital admissions per 10-μg/m$^3$ increase in PM$_{2.5}$ by major disease category for study subgroups, on average across all cities.**
(DOCX)

**S7 Fig. Percent change in hospital admissions per 10-μg/m$^3$ increase in O$_3$ by major disease category for study subgroups, on average across all cities.**
(DOCX)

**S8 Fig. Exposure–response curves of the effects of PM$_{2.5}$ on hospital admissions by major disease category, on average across all cities.**
(DOCX)

**S9 Fig. Exposure–response curves of the effects of $O_3$ on hospital admissions by major disease category, on average across all cities.**
(DOCX)

**S10 Fig. Percent change in hospital admissions per 10-μg/m$^3$ increase in $PM_{2.5}$ by major disease category using alternative model specifications, on average across all cities.**
(DOCX)

**S11 Fig. Percent change in hospital admissions per 10-μg/m$^3$ increase in $O_3$ by major disease category using alternative model specifications, on average across all cities.**
(DOCX)

**S12 Fig. Percent change in hospital admissions per 10-μg/m$^3$ increase in $PM_{2.5}$ by major disease category in 6 regions of China, on average across all cities.**
(DOCX)

**S13 Fig. Percent change in hospital admissions per 10-μg/m$^3$ increase in $O_3$ by major disease category in 6 regions of China, on average across all cities.**
(DOCX)

**S1 STROBE Checklist. STROBE checklist.**
(DOC)

**S1 Table. City-specific population and coverage rates of class 3 hospitals and hospital beds by the Hospital Quality Monitoring System (HQMS) in 252 Chinese cities.**
(DOCX)

**S2 Table. Annual number of hospital admissions and average cost per hospitalization in China.**
(DOCX)

**S3 Table. Demographic characteristics of cause-specific hospital admissions in 252 Chinese cities, 2013–2017.**
(DOCX)

**S4 Table. Summary statistics of annual-average levels of ambient air pollutants and weather conditions in 252 Chinese cities, 2013–2017.**
(DOCX)

**S5 Table. Pearson correlation coefficients among daily levels of ambient air pollutants and weather conditions, on average across all cities.**
(DOCX)

**S6 Table. Percent change in hospital admissions per 10-μg/m$^3$ increase in $PM_{2.5}$ (lag 0 days) by minor disease category, on average across all cities.**
(DOCX)

**S7 Table. Percent change in hospital admissions per 10-μg/m$^3$ increase in $PM_{2.5}$ (lag 1 day) by minor disease category, on average across all cities.**
(DOCX)

**S8 Table. Percent change in hospital admissions per 10-μg/m$^3$ increase in $PM_{2.5}$ (lag 2 days) by minor disease category, on average across all cities.**
(DOCX)

**S9 Table. Percent change in hospital admissions per 10-μg/m$^3$ increase in PM$_{2.5}$ (lag 3 days) by minor disease category, on average across all cities.**
(DOCX)

**S10 Table. Percent change in hospital admissions per 10-μg/m$^3$ increase in PM$_{2.5}$ (lag 0–1 days) by minor disease category, on average across all cities.**
(DOCX)

**S11 Table. Percent change in hospital admissions per 10-μg/m$^3$ increase in PM$_{2.5}$ (lag 0–2 days) by minor disease category, on average across all cities.**
(DOCX)

**S12 Table. Percent change in hospital admissions per 10-μg/m$^3$ increase in PM$_{2.5}$ (lag 0–3 days) by minor disease category, on average across all cities.**
(DOCX)

**S13 Table. Percent change in hospital admissions per 10-μg/m$^3$ increase in O$_3$ (lag 0 days) by minor disease category, on average across all cities.**
(DOCX)

**S14 Table. Percent change in hospital admissions per 10-μg/m$^3$ increase in O$_3$ (lag 1 day) by minor disease category, on average across all cities.**
(DOCX)

**S15 Table. Percent change in hospital admissions per 10-μg/m$^3$ increase in O$_3$ (lag 2 days) by minor disease category, on average across all cities.**
(DOCX)

**S16 Table. Percent change in hospital admissions per 10-μg/m$^3$ increase in O$_3$ (lag 3 days) by minor disease category, on average across all cities.**
(DOCX)

**S17 Table. Percent change in hospital admissions per 10-μg/m$^3$ increase in O$_3$ (lag 0–1 days) by minor disease category, on average across all cities.**
(DOCX)

**S18 Table. Percent change in hospital admissions per 10-μg/m$^3$ increase in O$_3$ (lag 0–2 days) by minor disease category, on average across all cities.**
(DOCX)

**S19 Table. Percent change in hospital admissions per 10-μg/m$^3$ increase in O$_3$ (lag 0–3 days) by minor disease category, on average across all cities.**
(DOCX)

**S20 Table. Percent change in hospital admissions per 10-μg/m$^3$ increase in PM$_{2.5}$ and O$_3$ for 8 negative control outcomes, on average across all cities.**
(DOCX)

## Author Contributions

**Conceptualization:** Jiangshao Gu, Ting Chen.

**Data curation:** Jiangshao Gu, Ying Shi, Yifang Zhu, Ning Chen, Haibo Wang, Zongjiu Zhang, Ting Chen.

**Formal analysis:** Jiangshao Gu.

**Funding acquisition:** Jiangshao Gu, Ning Chen, Ting Chen.

**Investigation:** Jiangshao Gu.

**Methodology:** Jiangshao Gu.

**Project administration:** Jiangshao Gu.

**Resources:** Jiangshao Gu.

**Software:** Jiangshao Gu.

**Supervision:** Haibo Wang, Zongjiu Zhang, Ting Chen.

**Validation:** Jiangshao Gu.

**Visualization:** Jiangshao Gu.

**Writing – original draft:** Jiangshao Gu.

**Writing – review & editing:** Jiangshao Gu, Ying Shi, Yifang Zhu, Ning Chen, Haibo Wang, Zongjiu Zhang, Ting Chen.

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
