## [Decision Letter · Decision Letter 0]

11 Feb 2020

Dear Dr. Gu,

Thank you very much for submitting your manuscript "Ambient air pollution and cause-specific risk of hospital admissions in China: A nationwide time-series study" (PMEDICINE-D-19-04229) for consideration at PLOS Medicine. 

[LINK]

In light of these reviews, I am afraid that we will not be able to accept the manuscript for publication in the journal in its current form, but we would like to consider a revised version that addresses the reviewers' and editors' comments. Obviously we cannot make any decision about publication until we have seen the revised manuscript and your response, and we plan to seek re-review by one or more of the reviewers. 

We expect to receive your revised manuscript by Mar 03 2020 11:59PM. Please email us (plosmedicine@plos.org) if you have any questions or concerns.

We look forward to receiving your revised manuscript. 

Sincerely,

Clare Stone, PhD

Managing Editor 

PLOS Medicine

plosmedicine.org

Abstract ‘risk of many diseases’ – would it be better to say non-communicable diseases? To be more specific; ‘2013 to 2017’ – please state the months; ‘Hospital Quality Monitoring System of China.’ – how many hospitals does this cover? And if it is only certain cities, please state which or at least say mostly in the East / West or whichever. Please add some summary demographic information to the abstract. 

Data – please state what the restrictions are. 

Methods – was written consent provided?

Table 1 – please remove roman numerals as not needed in a table

For non Chinese readers please classify what CCS is and their use

Please provide a table in the main text with demographic information of the participants.

I would remove ‘but also eliminate the publication bias that occurs when 340 multiple outcomes are examined and significant associations are selectively reported’

Did your study have a prospective protocol or analysis plan? Please state this (either way) early in the Methods section.

c) In either case, changes in the analysis—including those made in response to peer review comments—should be identified as such in the Methods section of the paper, with rationale.

Comments from the reviewers:

Reviewer #1: Gu and colleagues conducted a national time-series study on the association between air pollution and cause-specific hospital admissions in China. It is ambitious to include a vast number of cause-specific endpoints in one manuscript, but the rationale and biological plausibility should be better presented. The analytic approaches seems sound, and the draft is well prepared. Some of the major conclusions have just been reported in several recent publications (e.g. Tian et al, BMJ 2019, Plos Med 2019). I have several major concerns in data source and methodology, which need to be clarified further. There are other minor points that need to be fixed or improved. My detailed comments are as follows:

1. Line 90, more information is needed for the data collection, i.e. is it a public available dataset that can be accessed and supervised? The website or URL of it? 

2. Line 97, how were the cases diagnosed in the first place? The diagnosed cause of diseases upon hospital admission may be from the discharge diagnosis. In addition, hospital admission is quite dependent on local medical resources (hospital beds, etc) and personal social economic status (health insurance for instance). These factors may introduce large uncertainty in the analysis and may eventually bias the results (such as lag pattern). The authors should add a through discussion on this and compare the situation between China and developed countries.

3. Line 102, my biggest concern for this analysis is the rationale and biological plausibility for the association between air pollution and the endpoints examined. Although the authors cited previous papers that examined the same endpoints or exposures, there's inadequate information provided on the specific mechanisms for all the major categories included in the analysis in both Introduction and Discussion. Please amend. Data availability and clinical meaningfulness is not a convincing objective.

4. Line 108, more information is need for the exposures. For example, is this based on fixed-site monitoring stations? How many monitors are there for each city? If such information is provided in the Appendix, please incorporate and indicate. How about missing rates? I assume that will be a lot of missing in exposure time-series, especially for the early years and the less-developed areas.

5. Line 116, how were the statistical models determined? If modeling specification is based on previous studies, please cite them and explain the rationale to general readers. Why were the over dispersed Poisson family used, rather than the quasi-Poisson conjunction? What models exactly? Is linear regression models used?

6. Line 118, how were the degree of freedom determined in the first place? By AIC, BIC or other model fitting statistics? 

7. Line 121, why two terms of temperature and relative humidity were added? I assume this would introduce large collinearity in the models and cause overfitting. Is the best-of-fit modelling strategy applied? What model-fitting statistics were used?

8. Line 129, the main lag should be determined by how the models are fitted, especially for multi-center studies, rather than based on which to generate the largest effect estimate.

9. Line 134, how were north and south regions separated in China, by latitude?

10. Line 137, there's too little information on how the exposure-response curves were pooled, please provide a brief introduction. In particular, did the authors use a linear model for plotting, how is non-linearity considered for the associations between exposure and outcome?

11. Line 138, please indicate what dfs were used for each parameters, and why did the authors select these dfs rather than other ranges? For example, why only 8 to 12 for year, while other studies have reported 4 to 12. This should be stated for temperature and relative humidity as well.

12. Line 140, in the disease burden calculation, one critical problem is that this was inherently a time-series study on the short-term associations, thus it is not applicable to estimate the annual (long-term) reduction in hospital admissions. The authors should better clarify this. How is the information acquired on the average cost for each hospitalization in China? I assume this would vary a lot by cities.

13. Line 165, why to select different lags for PM2.5 and ozone? How to interpret the results if different lags were set?

14. Line 282, why were the estimates of this study lower than those from developed countries?

15. Line 286, why could this study provide evidence for the chronic health effects of air pollution?

16. Line 293, Tian's study was on ER visits, thus may not be appropriate for this case.

17. Again, please add through discussions on the rationales between air pollution and various endpoints.

18. Line 337, the first strength of this study may stand only if there's solid supporting evidence for the hypothesized association.

19. Line 340, the generalizability of this study is largely dependent on whether the hospital admissions are representative enough for the local population. This should be mentioned before in the Methods, in that how many population (maybe in percentage) did the class 3 hospital serve? How many hospitals are there in each city, etc.

Reviewer #2: The manuscript describes a large multi-city time series study from China on air pollution exposure and numerous hospitalization (morbidity) endpoints. The study made use of national standardized reporting of hospitalization discharges. The data analysis methodology was generally state-of-the-art for multi-site time series and accounted for the multiple exploratory comparisons undertaken. Additional sensitivity analyses were important, although not as helpful in some respects as they might have been - see below. Finally, the manuscript was generally very well-written. I have some concerns, however.

Critique:

1. An important first question, given the plethora of published time series studies, including multi-city time series studies, some of which are from China, is whether yet another one is needed. This study has some features to recommend it. The large array of outcomes, while not without issues (see points 3-5, below), is unique, and the large number of hospitalization counts ensures adequate power for addressing many of the outcomes. There may be others that the authors want to highlight to address the specific contributions that justify yet another time series study. 

2. Although I may have missed it, there was no mention that the hospitalizations included only those for unscheduled urgent admissions. Scheduled hospitalizations, such as those for "maintenance chemotherapy/radiotherapy," which interestingly showed an association with PM2.5, should not be included in this analysis. 

3. The authors argue that including the large array of hospitalization endpoints was a strength of the study, and have addressed concerns about multiple comparisons. However, with such a large array of outcomes showing statistically significant associations, the issue of biologic plausibility becomes a more serious concern. Interestingly, the authors raise the issue of negative controls when they note: "... negative controls such as female infertility, which has no plausible relation to air pollution" (line 280). One could argue, in fact, that many of the outcomes showing statistically significant associations could be characterized in the same way. For example, there would seem to be tenuous plausibility for calculus of the urinary tract, and certainly none for "maintenance chemotherapy/radiotherapy," as noted above. It's possible that all of the outcomes demonstrating significant effects in this analysis, as the authors suggest, are providing novel insights into the far more global impact of air pollution on health than previously thought, but this seems very unlikely. And while systemic inflammation and oxidative stress as a unifying "downstream" mechanism underlying all air pollution health effects covers a wide swath of diseases, there remain many outcomes for which even this is a tenuous argument, many that heretofore we might even have thought to include as negative controls. Although currently there is no solution to this dilemma, the authors should consider including some very convincing negative controls, such as accidents, for example, which were not included in the current analysis. 

4. One of the hazards of including as many endpoints as were included in this study is the possibility that there may have been a very small number of daily counts for many of the endpoints, and in some cases no counts in some of the cities. This would result in very uncertain estimates of effect in those cases in the first stage of the analysis. The meta-analytic step presumably somewhat accounts for this by giving less weight to endpoints in cities with uncertain estimates. It would nevertheless be helpful to see the distribution of daily counts of each of the endpoints (not merely IQR as in Table 2) as a supplemental table in order to assess the magnitude of this issue. 

5. A related issue is that the more specific the diagnostic categories that are used, the greater the likelihood of outcome misclassification. That is, while diagnosis coding accuracy for a general endpoint such as cardiovascular disease is good, diagnosis coding accuracy for more specific endpoints such as pulmonary heart disease or acute bronchitis, for example, are much more suspect. Because of the focus on a large array of very specific outcomes in this study, the likelihood of misclassification of the outcomes is great. The impact of such errors in the outcome when the outcome is binary would be to bias to the null. The authors recognize this, but perhaps somewhat more elaboration in the Discussion is needed. 

6. It is well known that reporting the effect estimate from the single day with the largest estimate of effect is biased and results in overestimation of the true effects. The current preferred approach is to employ an unconstrained distributed lag model to estimate the unbiased net cumulative effect over several days, an approach which is arguably also more in line with current pathophysiologic understanding. However, in the current setting, for PM2.5 at least, it seems the choice to report only current day effects is not an unreasonable choice given the day lag structure presented for many of the endpoints (Figure S2). This is not the case for ozone, however. Figure S3 shows that there is no clear single lag that stands out, in contrast to the picture with PM2.5. Even for respiratory disease, the primary condition for which effects of ozone were observed, the choice of lag is not clear - positive effects are observed for day lags 0 through 3 with that for lag 1 being the largest, and with day lag 3 being the longest lag reported. For circulatory disease, alternatively, the current day has the largest estimate of ozone effect. The justification for reporting only lag 1 effects for ozone is therefore suspect. The results for ozone, at least, are an example of why employing a distributed lag model is preferable. Reporting only day 1 lag effects for ozone and respiratory disease, the lag with the largest effect estimate (Figure S3), is not recommended. For most of the other endpoints, reporting only the lag 1 effects for ozone (Fig S3) is an odd choice.

7. The concentration-response function (CRF) plots (Figure 3) show 95% confidence bounds that are constrained to have bounds of zero at the point where the endpoint and the PM2.5 axes are both zero. This makes little sense given that there are likely no days with extremely low PM2.5 concentrations in Chinese cities. One expects the confidence bounds to be tightest around the median of the PM2.5 concentration distribution where most of the data lie. Admittedly, what the authors have shown is commonplace in the literature, but it's a practice that should be avoided because it is not informative. 

8. The authors have repeated (line 230) what is commonly stated in the literature regarding the absence of an apparent threshold concentration where no effects can be identified. While this may be the case for PM2.5 and for most of the endpoints, importantly it does not appear to be the case for ozone and respiratory disease (Figure S9), the endpoint of most relevance for ozone. A threshold in the range of 40-50 ug/m3 of ozone seems to be present. The conclusions need to therefore be qualified to reflect this.

9. The results of the several sensitivity analyses were reassuring. However, the sensitivity analysis relating to the degrees of freedom (df) for the time trend is not adequate. Here the authors have reported results of varying the time trend df in the range from 8 to 12/year. This range is far too narrow to assess sensitivity of results to varying df and to be assured that the chosen df are adequate. Clearly, using too few df runs the risk of not adequately removing time trends, while it has been argued that using too many df runs the risk of encroaching on the effect of the air pollutant. Given that the time scale of interest for pollutant effects is from one to a few days, with 12 df/year that corresponds to a knot for approximately each month, there seems little danger that the pollutant effect would be encroached upon by even more than 12 df. The authors should be encouraged to report results of a sensitivity analysis with time trend df ranging from 6 to 18/year.

Reviewer #3: See attachment

Michael Dewey

[LINK]

---

## [Decision Letter · Decision Letter 1]

2 Apr 2020

Dear Dr. Gu,

Thank you very much for re-submitting your manuscript "Ambient air pollution and cause-specific risk of hospital admissions in China: A nationwide time-series study" (PMEDICINE-D-19-04229R1) for review by PLOS Medicine.

I have discussed the paper with my colleagues and the academic editor and it was also seen again by reviewers. I am pleased to say that provided the remaining editorial and production issues are dealt with we are planning to accept the paper for publication in the journal.

[LINK]

We look forward to receiving the revised manuscript by Apr 09 2020 11:59PM. 

Sincerely,

Clare Stone, PhD

Managing Editor 

PLOS Medicine

plosmedicine.org

Requests from Editors:

At line 526 I think you mean to say risk of admission (rather "risk of disease")? Please amend.

Can you please check the given link for data as this doesn’t seem to retrieve the suggested link, for me at least ((https://spms.hqms.org.cn/). 

Please use the "Vancouver" style for reference formatting, and see our website for other reference guidelines https://journals.plos.org/plosmedicine/s/submission-guidelines#loc-references (specifically, no bold or ital font and only the first 6 authors names followed by et al)

Comments from Reviewers:

Reviewer #2: I am generally very pleased with the authors' thorough responses to review, the responsiveness to reviewers' suggestions and the revisions to the manuscript. My major issues have been addressed. A couple of comments:

1. I would have liked to have seen the results of an analysis employing a larger number of df/year for time trend, but the authors' response provided justification for not doing that and it seems unlikely that there would have been substantial changes in the effect estimates with less smoothing.

2. It's unfortunate that the authors were unable to exclude non-urgent admissions, but their response was reassuring. 

Minor:

Line 225. It's probably not appropriate to indicate "According to the suggestion of reviewers, ..." in the body of a revised manuscript, but perhaps that decision could be left to the editors. 

Sverre Vedal

Reviewer #3: The authors have addressed all my points.

Michael Dewey

[LINK]

---

## [Editor Report · Decision Letter 2]

8 Jul 2020

Dear Mr. Gu, 

On behalf of my colleagues and the academic editor, Dr. Yuming Guo, I am delighted to inform you that your manuscript entitled "Ambient air pollution and cause-specific risk of hospital admissions in China: A nationwide time-series study" (PMEDICINE-D-19-04229R2) has been accepted for publication in PLOS Medicine. 

PRODUCTION PROCESS

PRESS

PROFILE INFORMATION

Thank you again for submitting the manuscript to PLOS Medicine. We look forward to publishing it. 

Best wishes, 

Clare Stone, PhD

Managing Editor 

PLOS Medicine

plosmedicine.org